# Effect of a Recliner Chair with Rocking Motions on Sleep Efficiency

**DOI:** 10.3390/s21248214

**Published:** 2021-12-08

**Authors:** Suwhan Baek, Hyunsoo Yu, Jongryun Roh, Jungnyun Lee, Illsoo Sohn, Sayup Kim, Cheolsoo Park

**Affiliations:** 1Department of Computer engineering, Kwangwoon University, Seoul 01897, Korea; zhsjzhsj@gmail.com (S.B.); byeng3@kw.ac.kr (H.Y.); 2Digital Transformation RnD Department, Korea Institute of Industrial Technology, Ansan 15588, Korea; ssaccn@kitech.re.kr (J.R.); blueleen2@kitech.re.kr (J.L.); 3Department of Computer Science and Engineering, Seoul National University of Science and Technology, Seoul 01811, Korea; isohn@seoultech.ac.kr

**Keywords:** rocking motion, sleep quality, sleep staging, spindle, real world data, deep neural networks

## Abstract

In this study, we analyze the effect of a recliner chair with rocking motions on sleep quality of naps using automated sleep scoring and spindle detection models. The quality of sleep corresponding to the two rocking motions was measured quantitatively and qualitatively. For the quantitative evaluation, we conducted a sleep parameter analysis based on the results of the estimated sleep stages obtained on the brainwave and spindle estimation, and a sleep survey assessment from the participants was analyzed for the qualitative evaluation. The analysis showed that sleep in the recliner chair with rocking motions positively increased the duration of the spindles and deep sleep stage, resulting in improved sleep quality.

## 1. Introduction

Sleep plays a major role in maintaining an individual’s physical health, owing to its involvement in several physiological processes such as tissue healing and repair [1,2,3]. Moreover, several studies claimed that sleep could critically affect physical and cognitive performances, thereby encouraging researchers to investigate how to achieve effective rest and improved sleep quality in a limited amount of time. Improved sleep quality could result in having a healthier condition and better quality of life [4,5,6]. Specifically, good sleep quality would relieve the risk of acquiring insomnia [7,8,9,10], which could critically impair the immune function [11] and cause neurodegenerative disorders and cardiac diseases [12]. On the other hand, poor sleep quality is associated to coronary artery diseases in adults such as myocardial infarction [13,14,15,16] and affective states like depression and anxiety [17,18], which may affect an individual’s work productivity, social functioning, and safety. These results suggest that the sleep quality is as crucial as sleep quantity in ensuring life satisfaction [19], which is one of the core indicators of a healthy lifestyle.

The benefits of sleep on angle-adjustable recliner chairs have been reported several times. It has been reported that sleep in an angle-adjustable chair, such as a recliner, improved the sleep quality through the sleep at a higher angle of posture or the neck [20,21,22]. In particular, it has been proven that some sleep disorders could be medically improved in patients with sleep apnea [22,23,24,25] and nocturnal gastroesophageal reflux through sleep in a recliner [26]. In this paper, we propose an additional way to further improve the quality of sleep with providing a rocking motion to the recliner. This study introduces a new swinging mechanism with rocking motions, which improves the quality of a day-time sleep in a recliner. Therefore, a new swinging recliner chair is proposed to improve the nap environment with controlling the rocking and reclining functions.

In order to analyze the quality of sleep in the proposed recliner chair, we estimate five different sleep stages, Wake, NREM1 (N1), NREM2 (N2), NREM3 (N3), REM, using every 30 s epoch of the brain waves signal based on American Association of Sleep Medicine [27]. With this, various sleep parameters [28], including total sleep time (TST), total N1 sleep (TN1), total N2 sleep (TN2), total N3 sleep (TN3), total REM sleep (TR), sleep efficiency (SE), slow-wave sleep ratio (SWS%), and SWS to light sleep ratio (SL%), and quantitative comparisons of the overall sleep stages are investigated for the qualitative and quantitative measures of sleep quality based on the functions of the swinging chair. Additionally, we also investigate the spindle components in the recorded brain waves during the nap, which is a specific brain wave pattern occurs during sleep because of bursts of neural oscillatory activity by interplay of the thalamic reticular nucleus (TRN) and other thalamic nuclei [29,30,31]. Since it has been reported that spindle plays a major role in the benefits of memory enhancement and sensory shut down for reducing wake up [32,33,34,35], spindle element could be one of major elements to evaluate quality of sleep [29,36]. This study analyzes the quantitative growth of the overall spindles based on two types of spindles, namely the slow and fast spindles [37,38]. There are several conflicting reports regarding these two types of spindles in terms of their roles in various cognitive tasks. Since there seems to be only a consensus on the quantitative effects of spindles to sleep quality and not on their mechanisms and qualitative effects, this study only conducted the quantitative analysis on the patterns of the slow and fast spindles. A deep learning model is designed to estimate the sleep phases, which decide the multiple sleep parameters, and the frequency analysis is utilized to detect the spindle components.

The plan of the paper is as follows. First, we propose a novel experimental method that produces rocking motions in a recliner chair. Here, the characteristics and factors of the ‘swinging-chair’ and the descriptions of the participant and the experimental environment are provided. Then, we discuss the automatic sleep staging methods for the sleep quality evaluation, where the architecture of the deep neural networks used to classify the sleep stages is addressed.

## 2. Materials and Methods

### 2.1. Recliner Chair

A recliner chair, which reliably reproduces identical movements, was designed using a 200 W linear actuator motor that provides robust, steady, and repetitive swinging motions to analyze the nap quality in the swinging chair. A link fixation mechanism yields a specific trajectory with a ±5 cm margin of error depending on the experimental condition. The range of the swinging motion was fixed at 10 cm head-on for all conditions [39], and the acceleration during the swing was 0.3 m/s 
${}^{2}$
 based on the ISO reference [40]. The swing cycle of the recliner chair is once per 4 s, that is 0.25 Hz. These operations were controlled by a microprocessor and managed by continuous logging of the accelerator signals. In addition, it automatically calibrates each swinging cycle using the position sensitive device (PSD) sensor to continuously track and minimize the trajectory error and calibrate the error values caused by the swinging environment or motion artifacts of the participants. For the comfort of the user, the back of the chair was fixed to 14° (±2°) and the seat of the chair to 36° (±2°) [41], as shown in Figure 1. There are two types of rocking conditions, namely ‘Condition A’ and ‘Condition H.’ The chair creates a lateral pitch motion on the *y*-axis in Condition A, while the chair provides a back-and-forth rectilinear motion on the *x*-axis in Condition H. Moreover, ‘Condition B’ is considered as the control in this study, in which the chair does not move in any axis. Figure 2 shows the illustrations of the chair movements in all conditions.

### 2.2. Experimental Protocols

Fifteen participants (age mean = 25.375, SD ± 2.99 years old, all male) with no history of sleep disorders were recruited for the nap experiment. All participants napped on the recliner chair three times for three hours in all three conditions. They were asked to follow a strict sleep and diet regimen the day before the experiment. The average body mass index (BMI) of the participants is 23.01 ± 2.09, and the average Pittsburgh sleep quality index (PSQI) [42] is 5.625 ± 3.22. Moreover, we controlled the environmental elements such as sound intensity, room temperature, relative humidity, and luminosity to ensure that the sleep environment was controlled. Moreover, during the experiment, the luminosity was controlled to be lower than 10 lux [43] and the noise level of the room was 30 dB on average and lesser than 45 dB [44,45]. The room temperature was fixed at 23 °C [46], while the relative humidity was set at 45% [46] using an air conditioner. Furthermore, sleep scoring two-channel electroencephalography (EEG) signals [47,48,49] were recorded from the participants during their entire nap periods.

Two scalp electrodes on Fp1 and Fp2 shown in Figure 3 were attached as working electrodes referenced to the left earlobe (A2) and grounded to the right earlobe (A1) to obtain the EEG signals from the participants. In addition, the physical removal of the very top layer of the skin (i.e., stratum corneum) and other foreign substances on the epidermis was carried out before the attachment of the electrodes and their maximum impedance was limited under 5 k
Ω
. A bandpass filter [50] in the range between 0.5 and 55 Hz [51,52] was applied to the recorded EEG signals to eliminate unnecessarily ambient noise such as motion artifacts from the recliner chair, and a notch filter was used to remove 60 Hz power line noise [53]. Moreover, MP36 (BIOPAC Systems Inc, Goleta, CA, USA) and gUSBamp (Gtec, Albany, New York, USA) were utilized to record EEG signals at 500 and 600 Hz sampling rates, respectively. The amplifiers were operated at a sufficient distance from the chair to prevent any interference of electromagnetic noise generated from the actuator of the recliner chair. Each participant conducted the nap experiments of three conditions in randomized order. In addition, all participants had been instructed to nap under similar conditions as the experiment before the actual recording to avoid the first night effect [54]. The nap time was limited from 14:00 to 17:00 considering the circadian rhythm of the participants, and they were under strict sleep and dietary restrictions as previously mentioned. All participants were prohibited from consuming caffeine and other psychiatric drugs and were also provided diet and sleep schedules, as shown in Figure 4.

Moreover, the participants were subjected in a more controlled condition two hours before the experiment, that is, being banned from drinking caffeinated drinks [53] and performing rigorous activities [56] while being watched by the researchers to avoid any bias. In addition, a PSQI [42] survey was conducted before the experiment for a qualitative analysis of the sleep quality of the participants, including the amount, depth, and comfort of sleep over a month, providing a subjective assessment to determine the sleep disorders of the participants. The PSQI varies from 0 points to a maximum of 21, which is segmented into 7 levels according to the scoring method. A high PSQI score indicates poor quality of sleep, and a score of 5 or higher decides a poor sleeper, while less than 5 tells a good sleeper [42,57]. In this paper, based on the PSQI score, the subjects are divided into ’good sleeper’ groups and ’bad sleeper’ groups, and then it is investigated which group achieves significant sleep quality improvement on the proposed recliner. Moreover, the participants and their environments were continuously monitored using a web camera to observe any instability during the experiment. After the experiment, a subjective sleep quality survey [58] was conducted again for another qualitative analysis, including the time of sleep and numbers of waking time.

This experiment was approved by the Institutional Review Board of Kwangwoon University (IRB No. 7001546-20200728-HR(SB)-006-01).

### 2.3. Sleep Stage Estimation

The sleep quality could be estimated using various sleep parameters based on the sleep scoring results. In this study, the sleep stages were defined using an automatic machine learning algorithm with a significantly acceptable performance than that of a manual polysomnography (PSG) scoring that is commonly used by clinicians [59,60,61]. The neural network-based sleep scoring algorithm, DeepSleepNet [62], trained using the Wisconsin Sleep Cohort dataset [63,64], was applied to test the participants’ EEG signals. In this paper, since the quality of sleep is analyzed through the AASM [27] based five-stage sleep states, the proposed model classifies the possible Wake, NREM1, NREM2, NREM3, and REM sleep stages using the input EEG signals.

The model consists of two small and large convolution neural networks (CNN) [65,66,67,68], as shown in Figure 5, which are divided into two branches and then merged through a concatenation, where a small CNN architecture captures temporal properties and a large CNN architecture extracts frequency information for the sleep stage estimation [62]. The bi-dimensional long-short term memory (LSTM) [65,69,70] is utilized in the DeepSleepNet to exploit the information of both hidden states at time inferences ’t−1’ and ’t+1’ in the sequence.

As reported by Supratak, A. et al. [62], a two-step training process is required to estimate the sleep stages using the deep learning model, that is, a pre-training and a fine-tuning process. Before the pre-training procedure, oversampling and undersampling were conducted to solve the class-imbalance problem. The pre-training process does not train the entire model in Figure 5, but only trains the upper two convolution network layers. In this study, EEG signals of 129 participants from a Fp1 configuration recorded at 100 Hz from the Wisconsin Sleep Cohort dataset were used to train the model. The training was performed with 48,860 epochs for each stage, extracted through the undersampling based on the number of the N1 sleep stage, which has the smallest number of epochs among the five stages. When training two convolution network layers, an ADAM optimizer [71] with a batch size of 100 and a learning rate of 
$10^{-3}$
 were used.

In the fine-tuning process, randomly shuffled sequential data was utilized rather than the class-balanced dataset and the entire model depicted in Figure 5 was trained with applying two pre-trained CNN parameters to the CNN model. The optimizer used in this step is the ADAM optimizer, and the batch size is set to 10. In order to train the sequence residual learning part including the LSTM, its learning rate was set to 
$10^{-3}$
, while the learning rate was set to 
$10^{-4}$
 since the pre-trained parameters were applied to the two convolution network layers. For regularization, two techniques were applied; one is a dropout of neurons with 0.5 probability to prevent overfitting as shown in Figure 5, and the other is an L2 to prevent the occurrence of too large parameters in the model. This L2 weight decay is implemented during the training process of the two convolution network layers in order to prevent the overfitting due to the ambient noise or movement artifacts. However, it was not applied in the sequence residual learning because it could limit the model capabilities of learning the long-term dependency. The values of sleep parameters were calculated to analyze the quality of sleep based on the results of the five-sleep stage estimation using the model. Furthermore, a quantitative analysis of the sleep quality was conducted using the calculated sleep parameters.

### 2.4. Sleep Spindle Estimation

A sleep spindle is a unique characteristic that marks the beginning of an N2 sleep stage since it is used to define the transition from N1 to N2 stage. A sleep spindle is typically observed to have frequency components between 11–16 Hz, which is also known as a sigma wave. There has been evidences that certain characteristics of the spindle activity are linked to various cognitive and motor functions while its exact effects on the body are only speculated upon [32,72]. Various previous studies have suggested that increased spindle activities were related to both implicit and explicit memory consolidation [73,74,75,76,77], while others reported that spindle deficits were related to various neurological diseases, including autism [78,79] and Alzheimer’s disease [80,81].

The medical community considers the spindle identification by human experts as the gold standard in estimating the sleep spindle. However, there are several limitations with this method, including its unreliability in both intra-rater and inter-rater data [82,83] and its high cost. Therefore, there has been various attempts to create an automated sleep scorer that could reliably identify sleep stages and spindles [84,85,86].

Specifically, Lacourse et al. [60] proposed the A7 algorithm, which inspired the algorithm used in this study. The A7 algorithm tries to calculate the sleep spindle duration, oscillation frequency, amplitude of a spindle, and other properties.

On the other hand, the YASA algorithm [61,87] is more economical than the A7 algorithm because it uses only three parameters instead of four but performs in similar manner [88]. The first parameter is the relative power of the sigma band frequency components compared to the broader band components (0 to 30 Hz) that was extracted using the short-term Fourier transform (STFT) [89]. An increase of this parameter indicates that the power increase is dominant in the sigma band. Meanwhile, the second parameter is the correlation between the sigma band and the broad band components that was calculated using the Pearson correlation coefficient [90] and FIR filter [91]. Finally, the last parameter is the temporally changing root mean squared (RMS) value. The YASA algorithm can distinguish the sleep spindle components by monitoring the three parameters. Raphael et al. [87] reported 86.6% ± 6.2 accuracy and 78.5 ± 9.4 F1-score to detect the spindles, which is similar to the results of human experts [60,61,87]. Figure 6 shows an example of the detected spindle event using this algorithm in a 30 s epoch of the EEG signal.

### 2.5. Sleep Parameters

Various sleep parameters, including TIB (time in bed), which total time in bed while experiments, TST (total sleep time) and TSP (total sleep period), are calculated for the quantitative analysis of sleep quality. Specifically, TST is the sum of the total sleep length except wake among all sleep stages, while TSP is the amount of the actual time spent in the sleep cycles estimated by the duration between the beginning of the first N1 sleep stage and the wake. Other sleep parameters, including the parameters about deep sleep periods such as N3 and about the light sleep and wake periods such as N1, N2, and REM stages, were also investigated to achieve more accurate analysis. Additionally this study uses the sum of the N1 sleep stages (TN1), the sum of the N2 sleep stages (TN2), the sum of the N3 sleep stages (TN3), the sum of REM sleep stages (TR), and N1, N2, N3 and REM ratios to TST (N1%, N2%, SWS%, REM%), to analyze the sleep quality based on the absolute amount and the relative amount of sleep stages. Along with this quantitative analysis of the sleep quality, parameters related to the latency of the sleep stages, were also analyzed, where the sleep latencies to the light (N1 and N2), deep (N3) and REM sleep stages are investigated. For further analysis of the sleep quality, SL% (SWS ratio to the light sleep), SOL (Sleep on set latency) and SE (Sleep efficiency), are also calculated. SL% shows the amount of the N3 sleep compared with the light sleep (N1 and N2), SOL indicates the latency to fall asleep. SE represents the total sleep time like TST, but is normalized by the TIB value. These parameters are derived using the formula in Figure 7.

### 2.6. Self-Reported Sleep Quality Analysis

The self-reported sleep quality was investigated as a subjective and qualitative evaluation of the nap. All participants answered the questionnaire proposed by ÅKerstedt et al. [58] regarding the evaluation of their own sleep after naps, resulting in comprehensive results on the sleep quality by comparing the self-reported feedback with the EEG measurement. In this study, the qualitative sleep quality of the participants based on the self-reported evaluation time, awake time, and sleep latency parameters.

## 3. Results

### 3.1. Sleep Stage Automation Evaluation

The model used in the experiment needs to be re-evaluated using the EEG from Fp1-M1 and Fp2-M1 configurations since the simulations in the DeepSleepNet study [62] have been conducted on the EEG signals from Fpz-Pz and Cz-Oz channel configurations. In addition, the Wisconsin Sleep Cohort dataset [63,64] recorded with the Fp1-M1 and Fp2-M1 channel configurations was employed to train the DeepSleepNet model with the recliner channel configurations. The performance of DeepSleepNet trained using the Wisconsin Sleep Cohort dataset is shown in Table 1. Results show that the average accuracy of the 240 participants was 81.16%, which is a similar level of the accuracy estimated by medical experts [59]. However, N1 stage is poorly detected compared with the other stages that have been consistently addressed in the previous reports [62], and the N1 and N2 sleep stages are usually considered as one stage, ’Light sleep’ [27] in four-sleep stage taxonomy [92,93,94,95,96,97]. We also applied the four stage scheme with combining the N1 and N2 stage as one ’Light sleep’ to analyze the sleep quality in multiple approaches.

The trained deep neural network model predicts the sleep stages of the participants sleeping in the recliner with the designed rocking motions to evaluate their sleep qualities. Figure 8 displays the estimated sleep stages of one of the participants for a 3 h sleep. Each sleep stage is decided in every 30 s epoch.

### 3.2. Sleep Parameter Analysis

The estimated sleep stage lengths of all 15 participants corresponding to the different recliner conditions, as shown in Figure 9 and Table 2, were predicted using DeepSleepNet trained by the Wisconsin Sleep Cohort dataset. Since the number of participants were not big enough to determine whether the data was fit for parametric tests, although the data mostly assumed a normal distribution, we utilized the Friedman’s two-way analysis, a non-parametric statistical test that detects differences between conditions with repeated measures. For post-hoc analysis, the Wilcoxon Sign test was administered.

The duration of N3 (deep sleep stage), which is measured in minutes, in Conditions A and H (*p* < 0.05 for Condition A and *p* < 0.01 for Condition H tested using Friedman two-way test) are significantly higher than in Condition B, which is the control of the study.

The parameters are separated into four categories, such as ’sleep stage (min)’, ’sleep stage (%)’, ’sleep latencies’, and ’sleep indexes’. The sleep stage (min) is the length of each sleep stage, sleep stage (%) is the percentage of each sleep stage length relative to the total sleep length, sleep latencies is the time length entering a particular sleep stage, and sleep indexes are the sleep parameters suggested by the American Association of Sleep Medicine (AASM) [27].

The rocking motions in condition H significantly shorten the length of light sleep stages shown in Figure 9 based on TN1, TN2, N1%, and N2%, which indicate light sleep. Meanwhile, the deep sleep-related parameters, including TN3 and SWS%, indicate the increase of the deep sleep stage of participants in Conditions A and H compared to Condition B. However, the length of the REM sleep stage has decreased significantly in Condition A, indicating a possible deterioration of sleep quality of the participants. For further investigation of these seemingly contradicting results, we discuss an in-depth analysis of participants’ sleep spindles, which play a bigger role in several other cognitive functions such as visual attention, selective attention, and impulse control based on various findings.

In addition, ‘sleep latencies (min)’, which is a major parameter in evaluating sleep quality [28], shows significantly shorter periods to get to the NREM3 stage compared to that of Condition B. However, the TST value, which is the indicator of the total amount of sleep, shows that the amount of sleep at rocking motion has decreased relatively, which is evident in Condition H. It was statistically significant that the overall amount of time and number of events about the N3 stage was increased compared to the control group.

### 3.3. Sleep Spindle Analysis

Sleep spindles in EEG signals were investigated as another index in assessing the sleep quality without the sleep scoring information. The spindle can be analyzed based on the total amount of time, the number of spindle events, and its density and duration. The results of the spindle analysis corresponding to each rocking motion condition are shown in Table 3.

Looking at the average spindle frequencies in each condition, the quantity of sleep spindles significantly increases in Condition A. Moreover, even considering the total spindle time and its density, that is, the average recurrence of spindles in a 30 s period, Condition A shows more promising results. However, the sleep spindles have overall not improved for participants in Condition H.

### 3.4. Qualitative Sleep Analysis

The results of the qualitative sleep analysis of the participants based on the self-reported survey are shown in Table 4. Results show that there were no significant difference among the rocking condition experiments based on the self-reported responses. Most of the participants reported similar sleep and awake time in all three conditions and felt longer periods of sleep than the actual ones measured qualitatively using their EEG signals. The discrepancy between the quantitative and qualitative assessments of the sleep quality has been reported in several previous studies [98,99].

PSQI scores tell that assess participants’ quality of sleep over one-month period, containing questions about their usual sleep habits, with scores ranging from 0 to 21. Using the PSQI scores, participants are differentiated into two groups: one with poor sleep habits and another with good sleep habits. As suggested by Buysse et al. [57], we were set a threshold, 5, to separate ‘Good Sleepers’ having scores points under 5 and ‘Bad Sleepers’ over 5. Based on this criterion, the sleep parameters of the ‘good’ and ‘bad’ sleepers in three different experimental conditions were looked into, which are summarized in Table 5. SWS%, TN3, and SL% are the indicators of deep sleep and, the ‘Bad Sleeper’ group shows a significant increase of these parameters than the ‘Good Sleepers’ group. Particularly, regarding the parameters most closely related to the sleep quality, SWS% and TN3, the ‘Bad Sleeper’ group has a significant increase. The self-reported sleep quality in Table 6 has no significant difference between the ‘Good Sleeper’ and the ‘Bad Sleeper’ groups. However, when comparing the sleep times between the groups, the ‘Bad Sleeper’ group has more improved results by the rocking motions than the ‘Good Sleeper’ group.

## 4. Discussion

As we can see in Table 2, there were significant increase of the amount of deep sleep (N3) in condition A and H. In addition, the significant decrease of N1 stage is confirmed in condition H. The decrease of light sleep and the increase of deep sleep could infer the improvement of sleep quality [28]. The decrease of sleep latency to deep stage in condition A (see Table 2), indicates that the rocking motions could help extend the period of the deep sleep stage by reducing the time to reach the deep sleep phase. These results demonstrated that the rocking motions of the recliner could enhance the length of the deep sleep stage. The results are consistent with the previous reports by Omlin et al. [39] and Perrault et al. [100], demonstrating that the rocking motion increased the proportion of deep sleep stages among all-night sleep stages. Thus, this result proved that the deep latency caused by the rocking motion was significantly decreased during the three hours of the short nap on the recliner. Several previous studies reported that increasing the length of deep sleep stages, which is one of the key indicators of the improvement of sleep quality denoted by TN3, is effective in regenerating the body [101,102,103] and strengthening immunity [104,105,106]. In addition, the investigation of the sleep parameters is considered as an additional quantitative sleep quality analysis corresponding to the three different recliner conditions shown in Table 2. Furthermore, we conducted the analysis the sleep quality using the 11 sleep parameters listed in Figure 7 for a more accurate quantitative analysis. The results of the quantitative analysis based on the sleep parameters in determining sleep quality using the automated sleep scoring algorithm are shown in Table 2.

A separate sleep analysis was conducted from the participants who were relatively exposed to the risk of sleep disorders with low sleep quality based on PSQI scores in Table 6. As a result, more significant improvement of N3 sleep stage (deep sleep) could be confirmed in the participant group with low sleep quality, and Table 6 also demonstrated that the improvement in an increase of N3 sleep stage in the ’Bad Sleeper’ group was bigger than that in the ’Good Sleeper’ group. Thus, it confirmed that sleep in the recliner with a rocking motion such as Condition H was more effective in improving the quality of sleep for the people who have poor sleep habits. Therefore, the rocking motions proposed in this paper could help the people with high PSQI scores to induce high quality of their sleep. In addition, for those with a relatively high risk of sleep disorders, it would have a positive impact on the improvement of the sleep quality with the proposed rocking motion. Previous studies reported that those with high PSQI scores (‘Bad Sleeper’) are more likely to develop depressive symptoms and might be more susceptible to dementia and other cognitive impairments [107,108,109,110]. Therefore, we suggest that more sleep in the rocking motion might prevent these diseases.

In this study, sleep experiments and its analysis with sleep stage and spindles were conducted only using the two-channel brain waves, which is different from the conventional polysomnography (PSG) method that uses multimodal sensors such as multichannel EEG, electromyogram (EMG), electrocardiogram (ECG), and respiration to score the sleep stages. Despite that, there were multiple studies that produced significant and robust sleep scoring results [111]. Moreover, several studies have demonstrated that a generalized sleep scoring model based on a single or multi-channel brainwave produced similar sleep stages as those by sleep experts [60,61,87].

The substantial long-term memory conversion and task memory enhancement during the spindle event have been reported previously [76,112]. Omlin et al. [39] and Perrault et al. [100] demonstrated that the rocking motion induced the enhancement of spindles. In Table 3, the sleep spindle analysis using the EEG signals also confirms that the rocking motion of Condition A induces more spindle occurrences and increases its amount during the nap. Since this increase of the spindle is known to reinforce positive sleep effects, such as long-term memory conversion for learning and memory [32,33], motor ability [113,114] and sensory shut down for reducing wake up [34,35], it could be suggested that the sleep in Condition H would significantly improved the mental ability. Especially, considering the previous studies reporting that the spindle could have a positive effect on implicit and explicit memory consolidation [73,74,75,76,77], it is expected that the recliner chair with the suggested rocking motions could improve the performance of the brain. However, the memory test of the participant after the nap was not included since the enhanced spindles by the rocking motion during the nap were not expected when designing the experiment initially. In addition, Table 3 shows the amplitude and frequency of the two different types of spindles, fast and slow spindle [38], where each of the two different types of spindles was consistently estimated from the reliable EEG signals based on the corresponding standard range of two spindles [33,37,38]. In the following studies, the effect of the rocking motions on the memory enhancement will be mainly investigated. Furthermore, multi-channel EEG signals will be recorded to apply the multivariate algorithms in order to extract the spindle component accurately, which could be blind source separation algorithms [115,116,117].

In Table 4, we conducted an analysis of qualitative sleep quality based on a survey from the participants. Unlike the quantitative sleep analysis, the results demonstrated that there was no clear difference among the subjects. The difference between these qualitative and quantitative sleep analysis results in evaluating the sleep quality has been reported several times in the other studies [118,119,120,121,122]. Considering the environment of this experiment might be unfamiliar to the participants even though the first night sleep effect [54] was considered and that the sleep duration of the participants was limited during the experiment, the subjective feedback from the participants in a qualitative level might have been affected.

## 5. Conclusions

In this study, we investigated the influence of rocking motions on sleep quality during a nap. The sleep parameter analysis has proved that rocking motions improve the sleep quality by increasing the amount of N3 (Deep Sleep) sleep stage, and especially in the proposed Condition A the rocking motion provides a significantly positive effect on the spindle enhancement. These could be indicators of an efficient sleep experience even during the short period of nap. In addition, in an analysis based on the PSQI scores, it was confirmed that the Rocking motion improves the sleep quality more effectively for the ’Bad Sleeper’ group, who would be exposed to the risk of sleep disorders. Furthermore, these rocking motion paradigms could be extended to night-time sleep, which encompasses six hours of sleep, that keeps the body in good conditions.

## Figures and Tables

**Figure 1 sensors-21-08214-f001:**
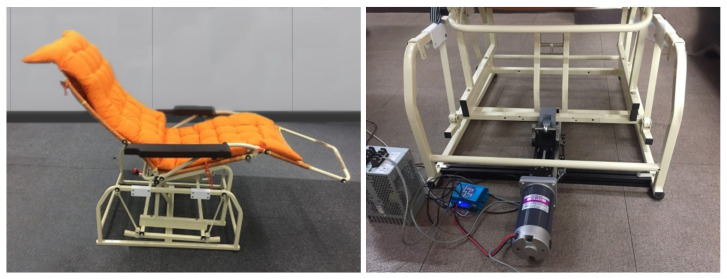
Designed recliner chair.

**Figure 2 sensors-21-08214-f002:**
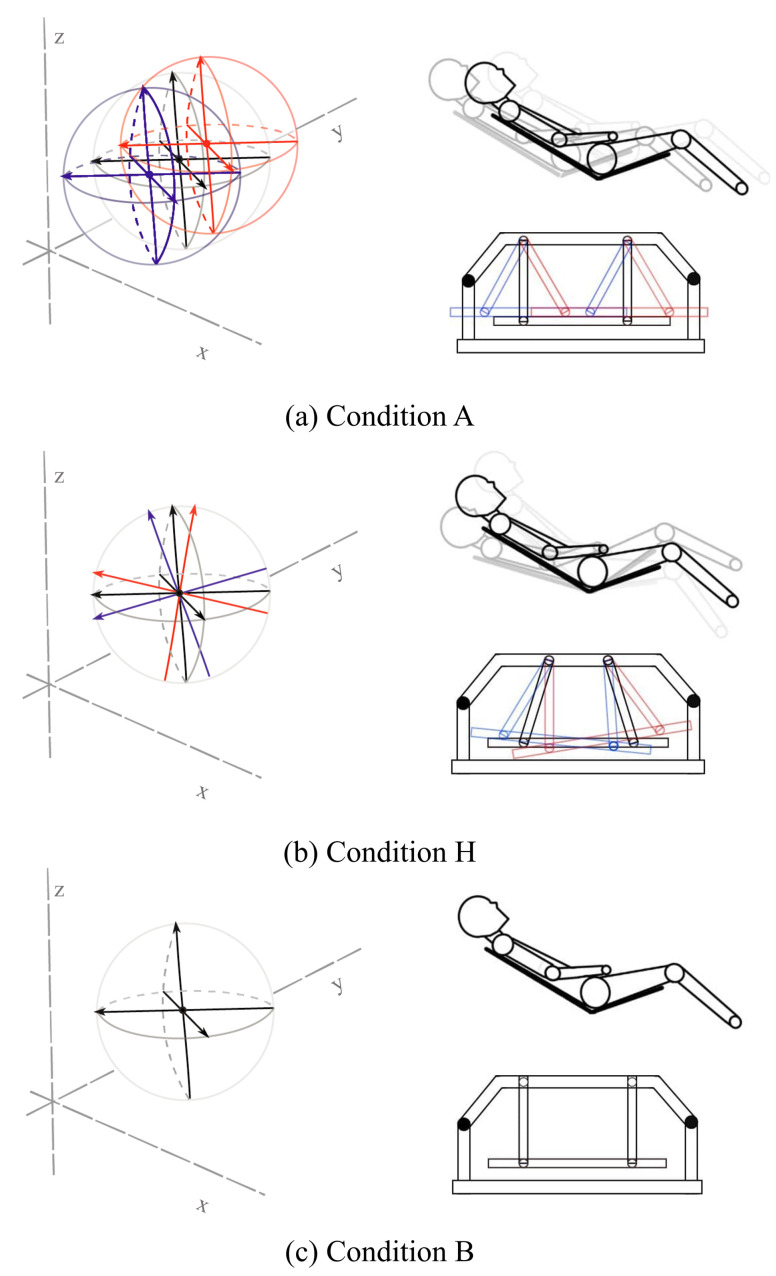
Rocking motion of the designed recliner chair. (**a**) Trajectory in the recliner Condition A. It has *y*-axis pitch movements; (**b**) Trajectory in the recliner Condition H. It has *y*-axis back-and-forth movements; (**c**) The control group is Condition B. It has no movement during the experiment.

**Figure 3 sensors-21-08214-f003:**
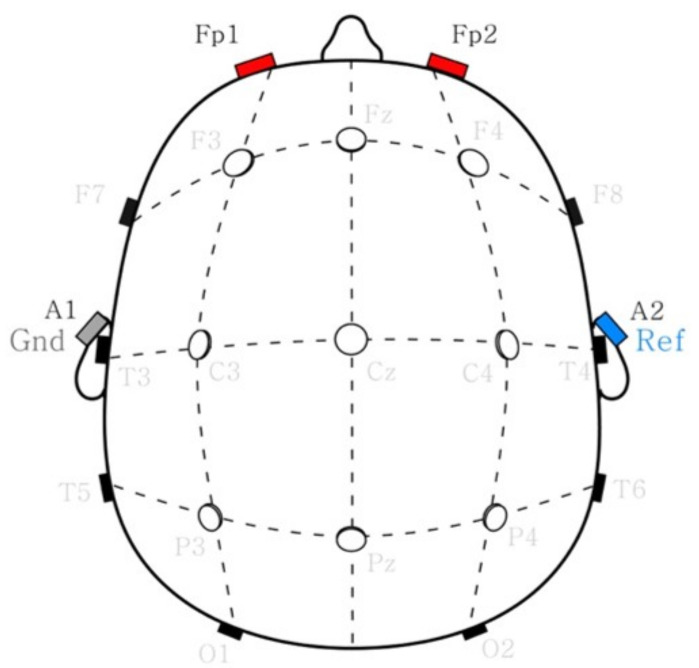
EEG montage of 10–20 systems [55], where Fp1 and Fp2 were used working electrodes referenced to A2 and grounded to A1.

**Figure 4 sensors-21-08214-f004:**
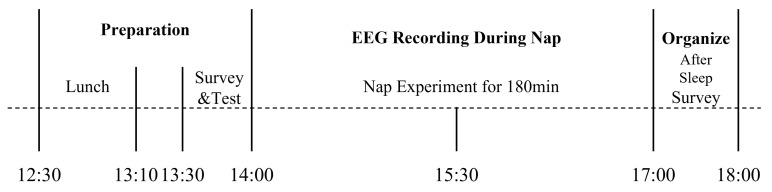
Experimental procedure to record EEG signals during the nap on the recliner chair for measuring nap time EEG.

**Figure 5 sensors-21-08214-f005:**
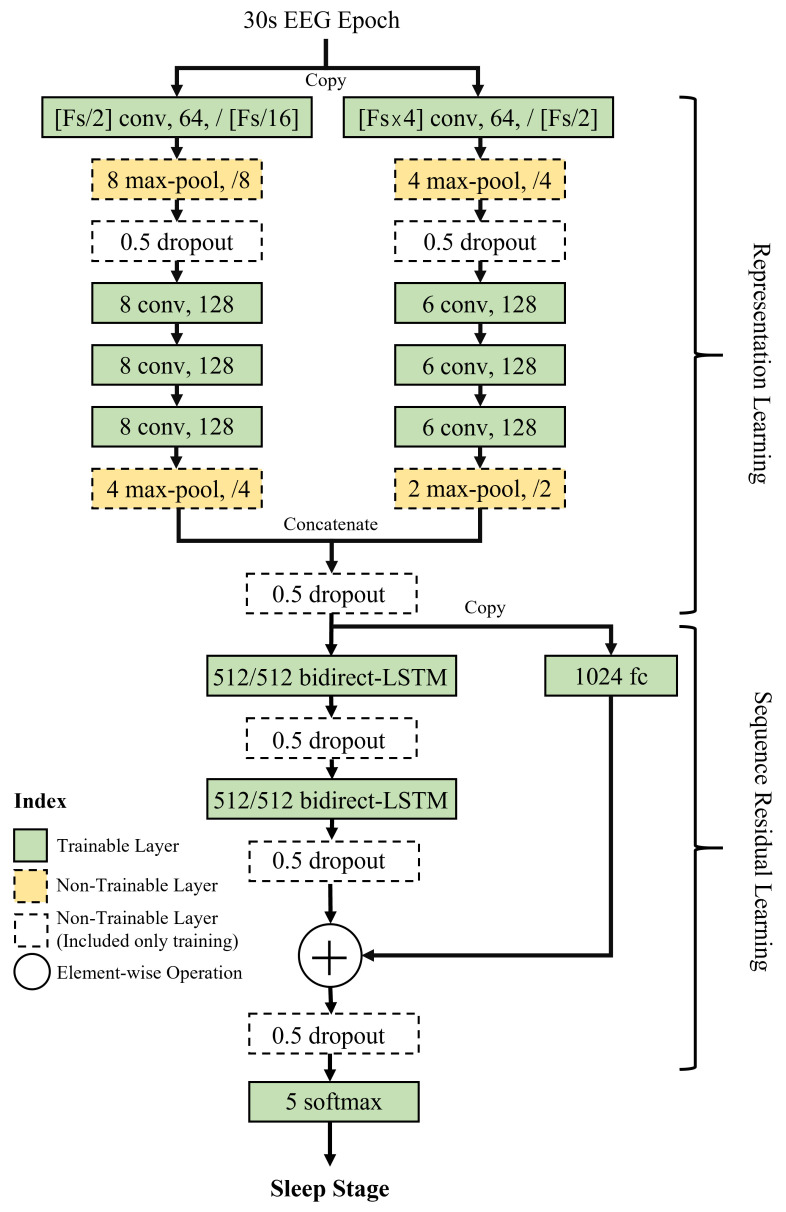
Architecture of the DeepSleepNet. It has two different learning parts such as the CNN and LSTM. The CNN layer proceeds the representation learning and LSTM proceeds the sequence residual learning. The EEG sampling rate (Fs) is a specification for the first convolutional layer.This model classifies the five sleep stages, defined by AASM [27] using the 30 s EEG epochs.

**Figure 6 sensors-21-08214-f006:**
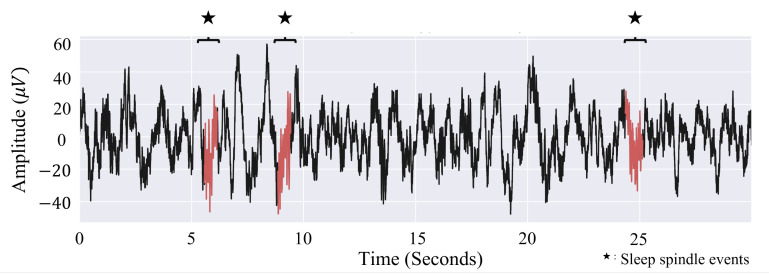
EEG spindle detected by YASA spindle detection algorithm. The red lines represent the detected spindle.

**Figure 7 sensors-21-08214-f007:**
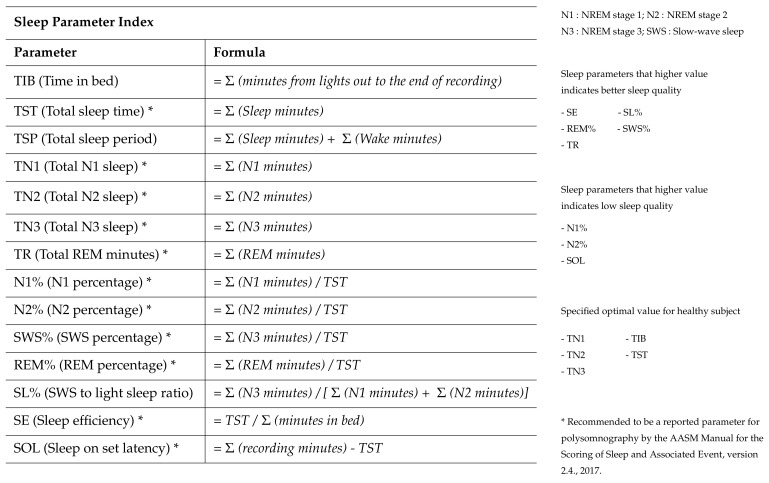
Sleep parameter index and its simplified formula.

**Figure 8 sensors-21-08214-f008:**
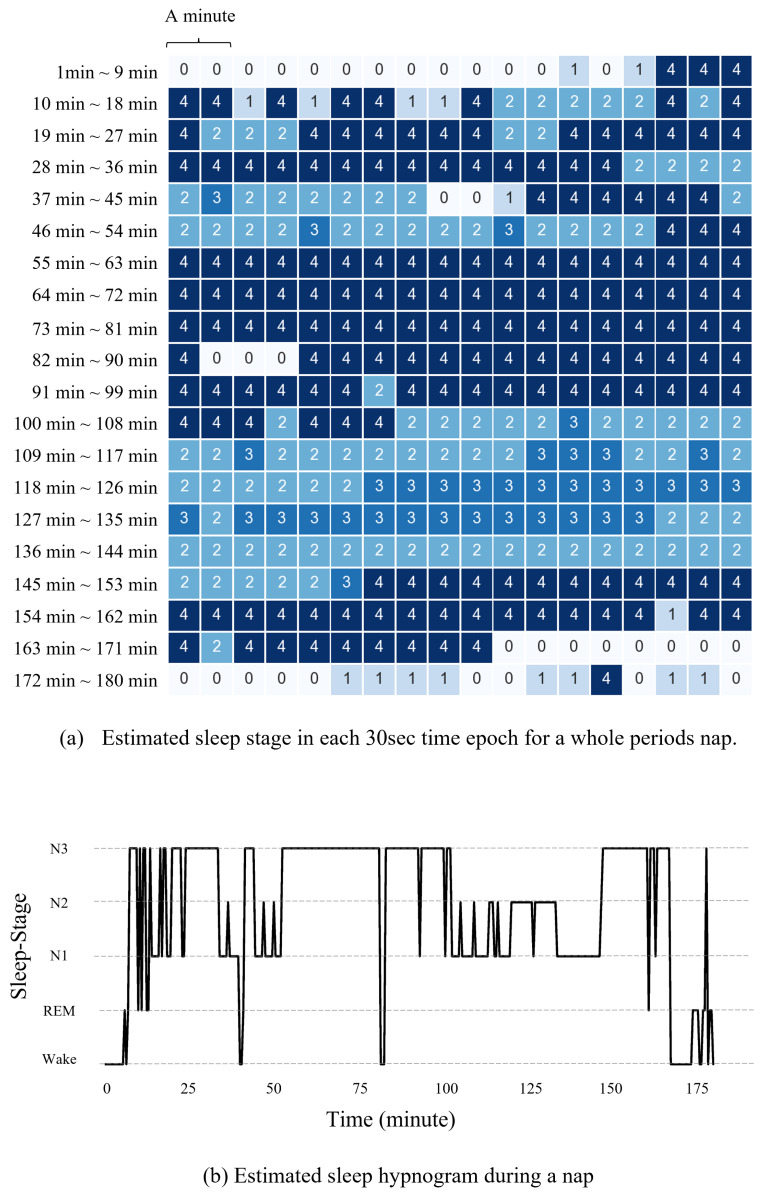
Estimaed hypnogram of one of the participants based on the sleep staging model. The estimation model architected by DeepSleepNet and trained by Wisconsin Sleep Cohort dataset. In (**a**), 0 is Wake, 1 is REM, 2 is N1, 3 is N2 and 4 is N3 sleep stage, respectively.

**Figure 9 sensors-21-08214-f009:**
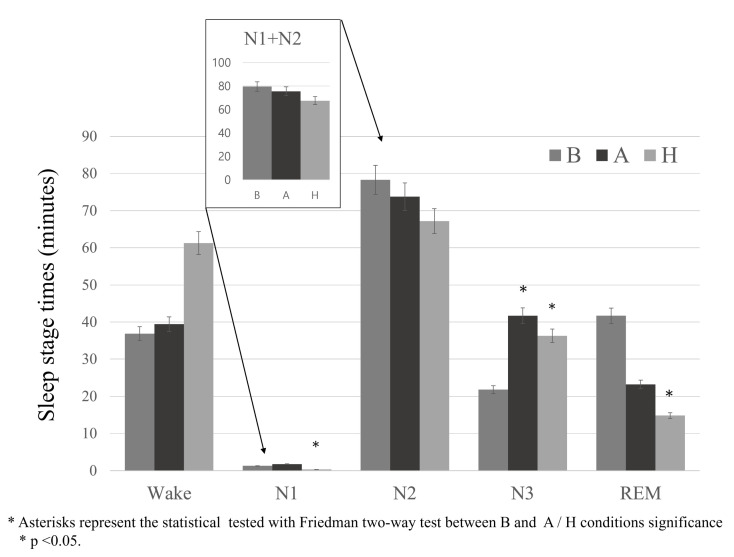
Estimated each sleep stage lengths of 15 participants during proposed nap experiments in the recliner chair.

**Table 1 sensors-21-08214-t001:** Evaluation of DeepSleepNet using the Wisconsin Sleep Cohort dataset.

	Wake (%)	N1 (%)	N2 (%)	N3 (%)	REM (%)
**Accuracy**	94.87 (±4.93)	92.24 (±4.43)	88.20 (±5.59)	95.31 (±3.30)	93.42 (±3.43)
**Precision**	85.00 (±12.37)	46.25 (±14.98)	80.64 (±13.96)	93.00 (±6.34)	85.10 (±16.50)
**Recall**	89.15 (±14.08)	48.20 (±18.08)	89.70 (±5.86)	76.35 (±14.51)	81.70 (±12.58)
**F1 score**	85.60 (±10.54)	44.00 (±12.90)	84.45 (±9.43)	83.05 (±9.44)	81.40 (±12.26)

**Table 2 sensors-21-08214-t002:** Sleep parameters in each experimental condition for the whole nap time of sleep analysis; Time in Bed is calculated from light-off; TSP, TST, and sleep latencies are calculated from the first N1 period (>1 min).

	B Condition	A Condition	H Condition
TIB [min]	180	180	180
TSP [min]	171.4 (±11.74)	172.67 (±7.63)	177.47 (±4.55)
TST ^★^ [min]	143.1 (±42.40)	140.53(±25.2)	118.7 (±42.50)
**Sleep Length [min]**			
TN1 (N1)^★^	1.3(±0.99)	1.77(±2.82)	**0.33(±0.48) ***
TN2 (N2)^★^	78.3(±36.42)	73.8(±29.92)	67.2(±31.57)
TN3 (N3)^★^	21.8(±20.07)	**41.73(21.51) ***	**36.33(±15.20) ***
TN1+TN2 (Light)	79.60(±36.37)	75.56(±29.95)	67.53(±37.76)
TR (REM)^★^	41.70(±43.14)	23.23(±34.66)	**14.83(±29.91) ***
**Sleep Ratio [%]**			
N1% (N1) ^★^	0.89(±0.69)	1.19(±1.71)	**0.24(±0.40) ****
N2% (N2)^★^	55.17(±25.02)	53.26(±20.26)	56.40(±15.19)
SWS% (N3) ^★^	18.46(±20.52)	**30.58(±15.85) ***	**34.08(±15.19) ****
N1%+N2% (Light)	56.05(±25.08)	54.45(±20.02)	56.64(±15.28)
REM% (REM) ^★^	25.48(±24.57)	14.97 (±20.44)	**9.27(±16.30) ***
**Sleep Latencies [min]**			
To Light (N1+N2)	21.70(±41.72)	21.00(±9.79)	35.60(±15.46)
To Deep (N3)	44.43(±23.85)	**21.03(±10.10) ***	30.23(±30.58)
To REM	16.56(±31.43)	25.53(±43.35)	27.00(±47.52)
**Sleep Indices**			
SL% [%]	57.54(±92.86)	72.30(±69.66)	69.15(±48.79)
SOL ^★^ [min]	36.90(±42.40)	39.47(±25.2)	61.30(±42.50)
Sleep Efficiency (SE) ^★^ [%]	79.50(±23.55)	78.08(±14.15)	65.94(±23.61)

★ Recommended to be a reported parameter for polysomnography by the AASM Manual for the Scoring of Sleep and Associated Event, version 2.4., 2017. * Asterisks represent the statistical tested with Friedman two-way test between B and A/H conditions significance * *p* < 0.05, and ** *p* <0.01.

**Table 3 sensors-21-08214-t003:** Sleep spindle analysis of EEG signals corresponding to each experimental condition during the three-hour nap.

	B Condition	A Condition	H Condition
Number of Spindle Event	77.43 (±50.76)	**113.99 (±66.30) ****	75.33 (±56.83)
Spindle Time (s)	68.32 (±45.00)	**96.22 (±56.00) ****	64.14 (±52.00)
Density (/30 s)	0.2673 (±0.0240)	**0.1879 (±0.0159) ****	0.1782 (±0.0207)
Duration (s)	0.83 (±0.28)	0.83 (±0.29)	0.83 (±0.30)
**Fast Spindles**			
Amplitude ( μ v)	56.29 (±35.50)	60.78 (±41.69)	65.18 (±59.76)
Frequency (Hz)	12.79 (±0.54)	12.79 (±0.60)	12.76 (±0.57)
**Slow Spindles**			
Amplitude ( μ v)	24.24 (±14.50)	29.41 (±25.60)	34.47 (±31.18)
Frequency (Hz)	8.79 (±0.17)	8.61 (±0.21)	8.81 (±0.23)

* Asterisks represent the statistical tested with Friedman two-way test between B and A/H conditions significance ** *p* <0.01.

**Table 4 sensors-21-08214-t004:** Self-reported sleep quality after the nap of the participants in three experimental conditions.

	B Condition	A Condition	H Condition
Time in Bed (min)	180	180	180
**Self-reported Survey Results**			
Sleep Latency (min)	13.13 (±10.29)	12.53 (±7.92)	9.60 (±5.78)
Wake Time (min)	30.67 (±19.22)	28.93 (±15.31)	31.87 (±16.91)
Sleep Time (min)	149.33 (±19.22)	151.07 (±15.31)	148.13 (±16.91)
SE (Sleep Efficiency)	0.83 (±0.11)	0.84 (±0.06)	0.82 (±0.09)

**Table 5 sensors-21-08214-t005:** Comparison of the quantitative sleep parameters between two groups separated by PSQI levels ‘Good Sleeper’ and ‘Bad Sleeper’.

		B Condition	A Condition	H Condition
Good Sleeper	SL%	0.82 (±1.15)	0.60 (±0.49)	0.73 (±0.51)
(PSQI < 5)	SWS% ^★^	0.25 (±0.24)	0.28 (±0.12)	**0.35 (±0.15) ***
9 subject	TN3 (min) ^★^	27.72 (±23.56)	39.06 (±14.20)	37.22 (±18.08)
Bad Sleeper	SL%	0.21 (±0.17)	0.91 (±0.95)	0.61 (±0.50)
(PSQI ≥ 5)	SWS% ^★^	0.09 (±0.06)	**0.34 (±0.22) ***	**0.32 (±0.14) ***
6 subject	TN3 (min) ^★^	12.92 (±9.04)	**45.75 (±30.67) ***	**35.00 (±11.00) ***

★ Recommended to be a reported parameter for polysomnography by the AASM Manual for the Scoring of Sleep and Associated Event, version 2.4., 2017. * Asterisks represent the statistical tested with Friedman two-way test between B and A/H conditions significance * *p* <0.05.

**Table 6 sensors-21-08214-t006:** Comparison of the reported self-reported sleep quality between two sleep type groups ‘Good Sleeper’.

		B Condition	A Condition	H Condition
Good Sleeper (PSQI < 5)	Latency (min)	15.56 (±12.10)	15.33 (±9.14)	9.22 (±4.09)
9 subject	Sleep time (min)	154.44(±18.37)	148.22 (±17.09)	147.22 (±16.41)
Bad Sleeper (PSQI ≥ 5)	Latency (min)	9.50 (±5.96)	8.33 (±2.58)	10.17 (±8.13)
6 subject	Sleep time (min)	141.67 (±19.41)	155.33 (±12.36)	149.50 (±19.11)

## Data Availability

Not applicable.

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
