# Peer review of "Effect of a Recliner Chair with Rocking Motions on Sleep Efficiency"

_sensors, 2021, doi:10.3390/s21248214_

Round 1

Reviewer 1 Report

  1. My main concern is with how the DeepSleepNet algorithm is being trained for this study? If you look at the original paper for DeepSleepNet (Ref #41), there are many steps involved in the training of the algorithm such as pre-training, fine-tuning, regularization, etc. For each step, careful selection of parameters is required and at least a few different values should be tried to decide the best parameter. The authors have skipped all of the details regarding how the model was trained. Also,  2 EEG channels are available, but it is not clear from Table 2 whether the accuracies are reported for channel Fp1 or Fp2 or is the average?  
  2. For Figure 6, ideally belongs in the results section and not under methods. Also, it is not clear whether it was calculated using the default DeepSleepNet or the new model trained using Wisconsin Sleep Cohort?
  3. In Fig. 6a, the assignment of numbers 0 to 4 to the sleep stages N1-N3, REM, and wake is not clear. This should be mentioned in the caption.
  4. In Fig. 6b, check that the time scale is showing 360 minutes. That is impossible since the experiment lasted only 180 minutes. 
  5. EEG signals are easily contaminated by motion artifacts. How did you ensure that rocking motion of the chair did not introduce any motion artifacts or cable pulling?
  6. Page 8, lines 205-211 should be moved to discussion.
  7. What is the long-term implication or benefit of this study? Because as per the self reported sleep quality assessments, participants did not notice any difference in their naps under different conditions.

Reviewer 2 Report

The study “Effect of a Recliner chair with Rocking Motions on Sleep Efficiency ” is complete and interesting. The use of sleep scoring and spindle detection models seems to be effective for the evaluation of the improvement of the  sleep quality when using a recliner chair with Rocking Motions. However, the innovative contribution of the study is not clear.

It could benefit from a simple changing in the text as currently both results and discussion sections are not well reported and discussed. 
In the results section you should only report the values of your analyses, the argumentation of the results should be included in the discussion. I suggest you review the contents of Results and Discussion sections because you have often introduced parts in the results that should be included in the discussion.

LINE 205-214: It could be moved to the Discussion section.

LINE 230-231: It could be moved to the Discussion section. Lack of reference: there are square brackets without reference, is it a typo?

LINE 238: Lack of reference; there are square brackets without reference, is it a typo?

LINE 239-240 and LINE 243-244: They could be moved to the Discussion section.

LINE 253-258: It could be moved to the Introduction section.

LINE 296-305 and LINE 307-311: They could be moved to the Results section.

LINE 298-299: You wrote “we set a threshold, as 5 to define ‘good sleeper’ with those having points under 5 and ‘bad sleeper’ with points over 5. “ but in the Table 6 and Table 7 you classified Good Sleeper with PSQI>= 5 and Bad Sleeper with PSQI<5, are they typos?

Reviewer 3 Report

In this manuscript a new recliner chair, which allows two different rocking motions while sleeping, is proposed. The effects of the proposed rocking motions on quality of sleep are evaluated, both quantitatively and qualitatively.

The methods and the parameters employed for the sleep analysis are based on previously existing approaches. In my opinion, some issues have to be addressed before the paper can be considered for publication.

Major comments:

  • From the introduction section, even though it is clear to the reader the importance of the quality of sleep for life satisfaction, it does not clearly arise why a rocking chair should improve the quality of sleep. This aspect should be better deepened.
  • Starting from line 30, a series of parameters are listed, however it not clear what N1, N2, N3 refer to. Only later in the paper it comes to the reader that these are definition of sleeping stages. A brief introduction to the matter would improve the readability.
  • Sentence at line 36-37 does not make any sense since it does not provide a definition of spindles.
  • In paragraph 2.3 the machine learning algorithm to detect sleep stages is explained. However, it is not clear which are the sleep stages that should be identified. See comment on Introduction.
  • If in figure 6 the outcomes of the automatic classifier are reported as results of the experimental data acquired in this study, it should be moved in the Results section. Moreover, the correspondence between numbers and sleep stages is not clearly explained.
  • Lines 161-162: the sentence “The accuracy of the algorithm was found to have 74 % precision…” is wrong. Accuracy and precision are two distinct concepts, as you separated them in table 2. Please rephrase the sentence.
  • In paragraph 2.5, improve the description of the parameters. Not all the parameters listed in table 1 are described in the text.
  • I have some concerns about the statistical analysis employed. A dedicated short paragraph in Materials and Method section would be appreciated. Moreover, at line 203, it is stated that t-test was employed to compare condition A versus B and condition H versus B. Was the normality of data distribution checked? Otherwise, a non-parametric test would be more advisable, considering also the small sample size considered. In addition, data acquired in the different conditions A, H and B cannot be considered independent, since are tested on the same population. Therefore, an ANOVA analysis seems to be more appropriated to investigate the presence of difference among the three populations, rather than employing directly multiple t-tests. A further post-hoc analysis could serve for paired comparison of conditions A and H with respect to the B condition.
  • At line 226-227, it is stated that sleep time in stages N1 and N2 shorten with respect to baseline for both condition A and H. However, only condition H appears to show statistical significance for N1, please comment.
  • Moreover, in paragraph 3.1, it was stated that N1 and N2 stages were merged; still all the parameters are reported separately in the results. Clarify if they were merged, then parameters presented should follow.
  • In table 3, please report the units of measure for each group of parameters.
  • The Discussion should be revised. From line 294 to line 311 other results obtained from a classification of subjects in ‘good’ and ‘bad sleeper’ are reported. These parameters should be adequately introduced in the Methods section and results reported along the others in Result section. Please, in Discussion focus on results interpretation. Further discussion of the obtained results for sleep parameters and spindles analysis should be reported. In particular, at line 284, please specify for which set-up (condition A or H) the rocking motion actually induces more spindles occurrence.

Minor comments:

  • Figure 2: adding a reference system to the picture would improve the comprehension of the reader about the chair motions.
  • Add some references:

       Line 194: cite for the acceptable level of accuracy in sleeping stages       estimate.

       Line 231, 238: there are empty squared brackets left for references.

       Line 254: cite reports about the two types of spindles.

  • Figure 8: specify significance of asterisks in caption.
  • Revise for few typos, e.g., line 118 … was were trained …, line 167 …the for the…

Round 2

Reviewer 1 Report

No further comments

Reviewer 3 Report

The authors implemented all the revisions requested, addressing some methodological issues pointed out in the previous report. The paper structure improved as well. Therefore, even though the scientific soundeness fo the work is limited, in my opinion the paper can be considered for publication.
